# *Aedes aegypti* ecology and dengue infection in three agricultural areas of Côte d'Ivoire

Edwige M. A. Y. Kadjo[1,2]*, Sylla Yahaya[3], Négnorogo Guindo-Coulibaly[1],
Valery E. Adjogoua[3], Danielle D. Zoh[4], François D. Traoré[4], Fabrice K. Assouho[5],
Mardoché R. K. Azongnibo[6], Stéphane D. M. Kpan[1], Ahoua Yapi[1], Fabrice Chandre[7,8],
Maurice A. Adja[1,2]

1 Laboratoire de Biologie et Santé, UFR Biosciences, Université Félix Houphouët-Boigny, Abidjan,
Côte d'Ivoire, 2 Institut Pierre Richet, Institut National de la Santé Publique, Bouaké, Côte d'Ivoire,
3 Departement des Virus Epidémiques, Institut Pasteur de Côte d'Ivoire (IPCI), Abidjan, Côte d'Ivoire,
4 UFR des sciences et technologie, Université Alassane Ouattara, Bouaké, Côte d'Ivoire, 5 Departement
Ressources animales et halieutiques, UFR Agriculture, Ressources Halieutiques et Agro-Industrie,
Université de San Pédro, San Pédro, Côte d'Ivoire, 6 Institut de Géographie Tropicale, Université Félix
Houphouët-Boigny, Abidjan, Côte d'Ivoire, 7 MIVEGEC, UMR IRD-CNRS-Université de Montpellier,
Montpellier, France, 8 Institut de Recherche pour le Développement, Montpellier, France

* yapokadjo@gmail.com

## Abstract

### Background

Over the last ten years, there has been an upsurge of dengue outbreaks in Côte
d'Ivoire, with *Aedes aegypti* as the principal vector. The intensification of farming
activities and the living conditions of local populations could lead to a proliferation of
artificial breeding sites for *Ae. aegypti*, which would increase the risk of dengue trans-
mission in rural areas. The present study characterised the habitats of *Ae. aegypti*
larvae and assessed the risk of dengue transmission in three key agricultural areas of
Côte d'Ivoire.

### Methodology

*Aedes aegypti* (larvae, pupae and adults) were collected from human dwellings in
three key agricultural areas of Côte d'Ivoire during the rainy season. Risk indices
including traditional Stegomyia indices and pupal indices were estimated. RT-qPCR
was used to detect DENV in the pools of *Ae. aegypti*.

### Principal Findings

*Aedes aegypti* was the predominant species collected at the three locations. The
predominant breeding sites were discarded tanks in Songon-Agban and Tchanctévè,
and water storage tanks in Kaforo. High Stegomyia indices (house indices >5% and/
or Breteau indices >20) and pupal indices (PIH [0.8 - 2.2] and PIP [0.1 - 0.4]) were
recorded in all three sites, suggesting a high risk of dengue transmission. DENV-3

**Data availability statement:** All relevant data are in the manuscript and its supporting information files.

**Funding:** Research reported in this publication was supported by the Fondation Merieux under Award Number RESIS-ARBO1-FM-18 to AMA and Fond Solidarité Santé Navale (FSSN) Award Number RESIS-ARBO2-FSSN-21 to AMA. The content is solely the responsibility of the authors and does not represent the official views of the Fondation Merieux and Fond Solidarité Santé Navale (FSSN).The funders had no role in study design, data collection and analysis, decision to publish, or preparation of the manuscript.

**Competing interests:** The authors have declared that no competing interests exist.

was detected in 1/96 (3.6%) pools of *Ae. aegypti* collected as immature stages in Songon-Agban with a minimum infection rate (MIR) of 1.9 per 1000 mosquitoes.

## Conclusions/Significance

The findings of this study indicated a high entomological risk of dengue across the three agricultural sites. It is important that the potential for transovarial transmission of DENV-3 in *Ae. aegypti* is considered when formulating control strategies against this vector in Côte d'Ivoire.

## Author summary

Dengue has been identified as a significant public health concern in Côte d'Ivoire, with *Aedes aegypti* identified as the primary vector. The most effective method of controlling arboviral disease is to control the vector mosquitoes. These control measures have focused only on urban areas, particularly the city of Abidjan, and have not taken rural areas into account. The intensification of agricultural activities could lead to a proliferation of breeding sites for *Ae. aegypti*, which would increase the risk of dengue transmission in rural areas. The success of *Ae. aegypti* control at national level could be compromised if rural areas are excluded from the control effort. We studied the entomological risk of dengue in three key agricultural areas in Côte d'Ivoire. We found that *Ae. aegypti* larvae mainly colonised the water storage and discarded containers. The Stegomyia indices suggested a high risk of dengue transmission in these areas. The additional pupal indices suggested that the risk of dengue was higher in Songon-Agban. A pool of *Ae. aegypti* collected as larvae from this same site was positive for DENV-3. The findings of this study suggest that the current regulatory measures in place for water tanks and discarded containers in urban areas are inadequate for achieving national-level control of *Ae. aegypti*, as the same breeding sites persistently remain infested by this vector in agricultural areas. It is imperative that dengue control measures take into consideration the presence of *Ae. aegypti* in agricultural areas to ensure optimal outcomes.

## Introduction

*Aedes* mosquito species include vectors of viruses that present a substantial threat to global health and socio-economic stability [1]. Dengue viruses (DENV), yellow fever virus (YFV), chikungunya virus (CHIKV) and Zika virus (ZIKV) are the most notorious *Aedes* borne viruses due to the severity of illness and magnitude of epidemics [2]. Of these, dengue is the most widespread virus circulating in more than 100 countries in the World Health Organization (WHO) regions of Africa, the Americas, Eastern Mediterranean, South-East Asia and Western Pacific. The African continent has reported

increasing numbers of outbreaks [3–6]. In Côte d'Ivoire, a surge in *Aedes* borne arboviral diseases has been observed during the two last decades. After the first case of dengue virus serotype 3 in 2008 [7], other studies reported the circulation of serotype 1, 2 and 3 in the country [8–11].

Knowledge about *Ae. eagypti* ecology is necessary to understand the epidemiology of dengue and to plan effective vector control strategies [12]. In dengue-endemic countries, WHO recommends vector surveillance to, predict epidemics and evaluate control efficacy [13].

Entomological investigations carried out in Abidjan, the city most affected by dengue epidemics, have shown that *Ae. aegypti* is present and very abundant. This vector preferentially colonizes two categories of breeding sites, namely domestic water storage containers and discarded containers. The proliferation of these types of breeding sites is mainly due to the inadequate supply of drinking water and the poor management of waste [14,15]. In Africa, *Ae. aegypti* populations thrive in anthropogenic environments due to the availability of humans who serve as a blood source and human-made larval sites [16].

The economy of Côte d'Ivoire has long been based on agriculture. Many farms have been set up in rural areas of the country. The people who make up the workforce live in the villages within or close to the farms. Similar to some neighbourhoods in Abidjan, the majority of these village populations face inadequate water supplies and waste management from domestic and agricultural activities. This creates a diversity of breeding containers favourable to *Ae. aegypti.* Previous studies carried out on some industrial farms in the southeast of the country have shown an abundance of *Ae. aegypti* [17]. The villages located close to these farms increase the contacts between humans and vectors, leading to a higher risk of dengue transmission to these populations [18]. Because of these observations, we were interested in other agricultural areas in the country with a similar situation. In areas where vegetables, cotton and pineapples are cultivated, villages are embedded in or near the farms.

Two mechanisms of dengue virus transmission have been described in *Ae. aegypti*: horizontal and vertical transmission. With horizontal transmission, the virus is transmitted from mosquitoes to their vertebrate hosts. Vertical transmission is a mechanism in which the virus is transmitted by the females to their offspring [19,20]. The epidemiological impact of vertical transmission of dengue virus is still poorly understood but is thought to be important for the maintenance of the virus in mosquito populations during interepidemic periods [21,22].

Dengue virus has four distinct serotypes: DENV-1 to -4 [23]. In western Africa, DENV -1, -2 and -3 were isolated for the first time from human samples in Nigeria [24]. Several other dengue outbreaks have been reported in different countries, as for example Burkina Faso [25], Senegal [26] and Ghana [27]. In Côte d'Ivoire, epidemiological surveillance data of dengue has detected the circulation of serotypes 1–3 within human populations [7–11]. Also, previous studies have detected DENV-2 and -3 infections in *Ae. aegypti,* the main vector of these viruses in the country [28–30].

The frequency of vertical transmission of dengue viruses in *Ae. aegypti* populations remains an area of research that has yet to be fully explored and understood. In addition to the circulation of multiple serotypes in human populations and in the vector, the existence of a high frequency of vertical transmission could represent a major risk for the emergence of dengue epidemics. The present study characterised the habitats of *Ae. aegypti* larvae, estimated the prevalence of vertical transmission, and assessed the risk of dengue transmission in three major agricultural areas of Côte d'Ivoire.

## Methods

### Ethics statement

Prior to the initiation of the study, the study protocol was formally endorsed by the relevant local municipal authorities (ref: 047/CB.SG). Informed consent was obtained from local community leaders via oral means. The collection of mosquitoes in households was conducted with the explicit permission of the property owners and/or residents. It is noteworthy that the study did not involve the use of any endangered or protected species.

## Study areas

The study was conducted in three agricultural areas of Côte d'Ivoire: Songon-Agban (in southern), Tchancthévè (in southeastern) and Korhogo (in northern) (Fig 1). These villages are located within the confines of agricultural farms. The villages are situated in close proximity to the plantations, at an average distance of 100 metres. Songon-Agban is a village on the outskirts of Abidjan. This site is characterised by a subequatorial climate, with four seasons: a long rainy season (March-July), a short rainy season (September-November), a short dry season (August) and a long dry season (December - February). Its main activity is the production and sale of market garden produce grown on vast plains close to near the village. Tchancthévè is a village located in the commune of Bonoua, 60 kilometers from Abidjan. The same climate characterizes this village as Songon, both located in the south of Côte d'Ivoire. Since the 1950s, pineapple has been grown on large plots of land close to the village. In addition, there are also some oil palm and rubber plantations. Kaforo is a village in the Department of Korhogo, located 648 kilometers north from Abidjan. This village is characterised by a tropical humid climate, with one long rainy season (March-November) and a short dry season (December-February). It is an area where cotton has been grown for almost 40 years. The cultivated fields surround the village.

## Mapping and selection of houses surveyed

In each study site, a single entomological survey was conducted during the rainy season. The survey was carried out in October 2021 in Songon-Agban and Tchancthévè, and May 2022 in Kaforo. Two methods were used in this study: larval collection and aspiration captures of adults. A total of 325 houses were surveyed: 150 in Songon-Agban, 109 in

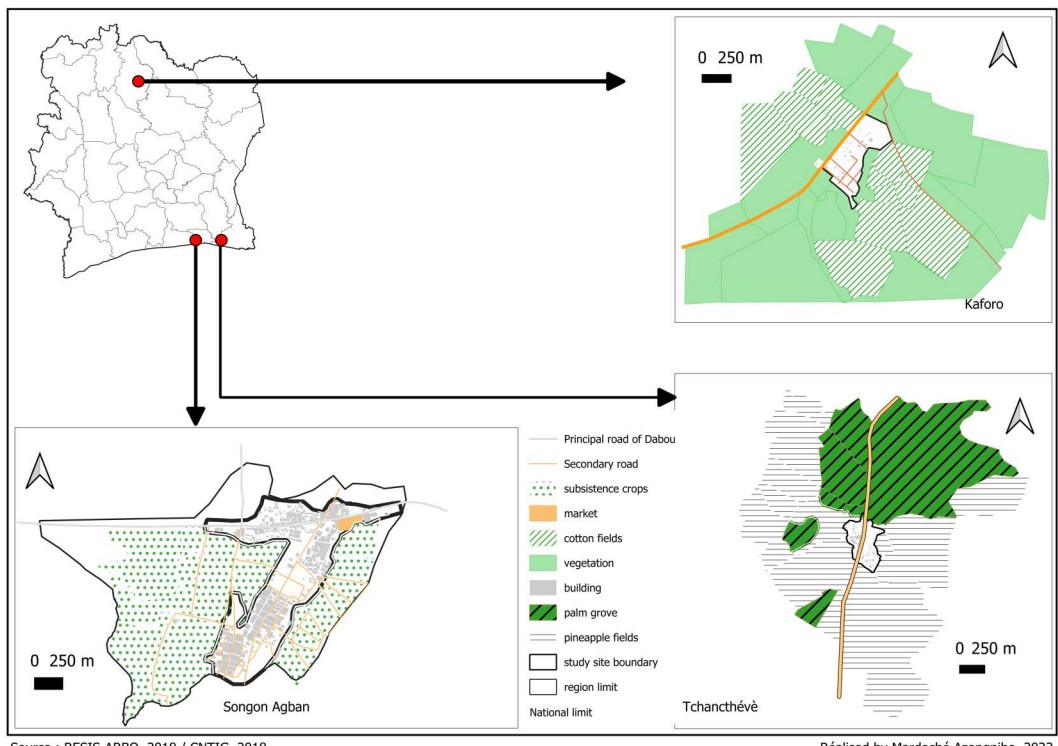

Source : RESIS-ARBO, 2018 / CNTIG, 2018                    Réalised by Mardoché Azongnibo, 2023

**Fig 1. Map of the 3 study sites.** The map was created with QGIS oftware version 3.34 (https://www.qgis.org/), using the basemap shapefile from the Database of Global Administrative Areas (GADM, https://gadm.org/; licence: https://gadm.org/license.html). The satellite images used to map the various cultivation areas were obtained using the HCMGIS plugin in QGIS (https://plugins.qgis.org/plugins/HCMGIS).

Tchancthévè and 66 in Kaforo. In Songon-Agban, the houses were randomly selected using a Geographic Information System (QGIS). The sample size of the 150 houses was determined using Yamane's simplified formula for calculating the sample size [31]. The following parameters were introduced into the Yamane's formula: 5% precision, 95% confidence level, an estimated of cluster in this site (N = 1570) and a response rate of 10%. The coordinates of the 150 houses retained in Songon-Agban were recovered in GPX format (GPS exchange Format) for transfer to a Garmin eTrex 20x global positioning system (GPS). For the other two sites (Tchancthévè and Kaforo), all houses were visited and georeferenced using a Garmin eTrex 20x (GPS). Before the selection of houses, these data were digitized using the OpenStreetMap platform. Entomological surveys were conducted at the houses indicated by the landmarks introduced into the GPS. At the Songon-Agban site, the Yamal formula applied to the sample population, which was the number of plots, was used to select 150 houses for the study. We applied random sampling using the QGIS random selection tool to select these 150 houses. If the inhabitants of a selected house refused to participate, the neighbouring house (left or right) was invited.

## Larval collection

The selected houses in each study area were visited. All domestic and peri-domestic containers containing water were inspected for the presence of mosquito larvae and pupae.The aim was to collect as many mosquitoes as possible for research into the vertical transmission of DENV. All larvae and pupae found in accessible containers were collected. In the case of large water storage containers, the containers were emptied with the consent of the residents so that all larvae and pupae could be collected. Otherwise, these containers were only counted as positive breeding sites. Very few of the natural breeding sites contained larvae or pupae. These were collected using a pipette. All natural and artificial water containers were defined as 'potential vector container' and those with immature stages (larvae and pupae) of *Ae. aegypti* as 'positive containers' [32]. Some data were noted on the containers: the type of container, the presence or absence of immature stages, the name of the sampling site, the house number and the number of residents. The containers were classified according to WHO guidelines used in dengue-endemic regions [33], namely water storage tanks or cisterns (>15 L); small water containers used for daily cooking and cleaning activities (<15 L); discarded tanks (including domestic waste and abandoned containers); natural containers (including natural tree, leaves of sheathed plants and bamboo holes); other containers (including wells, artificial containers used by households and which cannot be destroyed, e.g., animal drinking pots, flower pots) and used tires.

All immature stages of mosquito (larvae and pupae) were collected using a ladle or a pipette depending on the size of the container, from all positive containers. Samples were brought back in plastic cups to the insectarium of the Pierre Richet Institute to be reared until adult emergence. These larvae collected in the field were counted by genus (*Aedes, Culex, Anopheles* and *Toxorhynchites*) in each plastic cup. Those belonging to the genus *Toxorhynchites* were reared separately.

## Capture of adult mosquitoes by aspiration

Adult mosquitoes were colleted using a Prokopack aspiration (model 1412, John W. Hock Company, USA). All houses inspected for larval collection were also visited to capture adult mosquitoes. The captures were carried for 15 – 20 minutes in each house between 7–11 am and 3–6 pm corresponding to the periods of aggressiveness of *Aedes* [34,35].

The mosquitoes that emerged from the rearing of immature stages and those captured by aspiration were identified morphologically following the identification keys of Edwards [36] and Huang [37].

## Entomological indices

The level of transmission risk was estimated using standard Stegomyia indices including the Breteau Index (BI, the number of positive containers per 100 surveyed houses), House Index (HI, the percentage of houses infested),

and container index (CI, percentage of positive containers). The determination of these indices is recommended by WHO in dengue-endemic countries for the monitoring of vector populations to guide breeding site management, and evaluate control operations [13]. Additional pupal indices were calculated: the productivity index per house (PIH, the average number of pupae per house) and the productivity index per person (PIP, the average number of pupae per person related to the total persons leaving the household) [38]. These indices were calculated in addition to the larval indices because the density of pupal stages is considered to be a more reliable indicator of the abundance of adult mosquitoes [39].

### Detection of dengue virus in *Ae. aegypti* populations

**Preparation of Mosquitoes.** After morphological identification, *Ae. aegypti* females were pooled (up to 20 females per pool per method of collection) in 1.5 ml Eppendorf tubes containing 0.5 ml of RNA-Later, labelled, and stored in a freezer at -20°C. In total, 1753 female *Ae. aegypti* adults were used for DENV detection. These mosquitoes were composed of 119 females caught by Prokopack aspiration and 1634 females emerging from immature stages collected by larval surveys. The RT-qPCR was performed on 104 pools of *Ae. aegypti* females (8 from Prokopack aspiration and 96 from larvae collections).

**Extraction and amplification of viral RNA.** Nucleic acid (viral RNA) extraction was performed from 140 µL of mosquito pool ground supernatant using the QIAamp Viral RNA kit (Qiagen Catalog # 52904) according to the manufacturer's recommendations. The supernatant was obtained after grinding mosquito pool in phosphate buffered saline (PBS) and centrifugation at 8000 rpm for one minute. The different processes (amplification and detection) were carried out in the same reaction tube using the real-time RT-PCR monoplex developed by the Centers for Disease Control and Prevention (CDC) for the detection of dengue virus [40]. The PCR mixture (MasterMix) with a final volume of 25 µL was prepared with 12.5 µL 2X buffer + 8.25 µL Nuclease-free water + 1.25 µL of each primer (F and R) + 0.5 µL Probe + 0.25 µL RNAase (enzyme) + 1 µL RNA extract. PCR cycling conditions were reverse transcribed at 50°C for 10 min, initial denaturation at 95°C for 15 min, 40 cycles of extension at 95°C for 15s for denaturation, 95°C for 15s for the hybridation and 60°C for 1 min for the final elongation. The product was run on an Applied Biosystems LightCycler. The determination of the different dengue virus serotypes (DENV- 1, 2, 3 or 4) was carried out using the multiplex RT-qPCR [41]. The positive pools for the different viral serotypes were determined on the basis of the fluorescence emitted by the fluorophores, on an amplification curve of a real-time RT-PCR reaction.

**Percentage of pools testing positive (% Positivity) and minimum infection rate (MIR).** The % Positivity was estimated taking into account the total number of positive pools versus the total number of pools analysed and expressed as a percentage. The minimum infection rate (MIR) was calculated from the number of positive pools divided by the number of adults tested × 1000 [22,42,43].

### Statistical analysis

The evaluation of the risk of dengue transmission for each study site was estimated using the WHO criteria [44,45]. Whenever HI or CI or BI is < 5%, < 3% and 20, respectively, the area is considered unlikely to promote the transmission of dengue. Whereas if HI or CI or BI exceeds 5%, 3% and 20, respectively, an area is considered as high risk for *Ae. aegypti* to transmit dengue. Additionnally, the PIP values between 0.5 and 1.5 are considered indicative of dengue transmission risk [46]. Data were entered into a spreadsheet, cross-checked and transferred into GraphPad Prism version 5.0.1. The Pearson's Chi-squared test ($\chi^2$) was used to compare the proportion of positive breeding sites of *Ae. aegypti* populations and the larval indices (HI and CI) within the three study sites. The Kruskal-Wallis test (KW) was used to compare PPI values between these sites.

## Results

### Composition of Culicidae fauna of three agricultural areas

A total of 325 houses (150 in Songon-Agban, 109 in Tchancthévè and 66 in Kaforo), were surveyed during this study. From theses houses, 978 potential breeding containers for *Aedes* spp. were recorded, of which 201 (20.6%) were found to be positive for larvae and/or pupae of *Aedes*.The collected larvae supplied a total of 4545 adult mosquitoes that were represented four genera: 79.0% *Aedes*, 20.6% *Culex*, 0.3% *Toxorhynchites* and 0.1% *Anopheles*. *Aedes aegypti* accounted for more than 65% of the mosquitoes collected in each of the three study sites. A second species, *Ae. luteocephalus*, which also transmits dengue, were found at a lower proportion (0.1% for Songon-Agban and 1.1% for Tchancthévè). Concerning the genus *Culex*, *Cx. quinquefasciatus* was collected in all three study sites, and *Cx. nebulosus* only in Songon-Agban and Tchancthévè. A total of 318 mosquitos, representing three genera, were collected using Proko-pack aspiration. The *Aedes* genus, composed mostly of *Ae. aegypti,* was the most abundant of the fauna in the three study sites, followed by *Ae. luteocephalus* collected in low proportion only in Tchancthévè (Table 1).

### Types and prevalence of breeding containers of *Ae. aegypti*

Fig 2 illustrates the various breeding containers for *Aedes* mosquitoes identified during the survey. In Songon-Agban, 435 potential breeding containers were identified in the 150 houses visited. The potential breeding containers were comprised of 35.2% water storage tanks (153/435), 21.1% discarded tanks (92/435), 20.7% other containers (90/435), 16.3% small water containers (71/435), 6.2% used tires (27/435) and 0.5% natural containers (2/435). A total of 91 potential breeding containers (20.9%) were found infested with larvae and/or pupae of *Ae. aegypti*. These positive breeding containers consisted of 31.9% discarded tanks (29/91), 22.0% water storage tanks (20/91), 18.7% used tires (17/91), 15.4% other containers (14/91) and 12.1% small water containers (11/91). Of these, discarded tanks were the most frequently positive breeding containers ($\chi^2 = 43.48$; $p < 0.001$).

A total of 276 potential breeding containers were surveyed in the 109 houses in Tchancthévè. The survey identified several categories of potential breeding containers: water storage tanks (49.3%; 136/276), discarded tanks (19.2%; 53/276),

**Table 1. Composition of mosquito fauna collected from larvae and adult collections in three agricultural areas of Côte d'Ivoire.**

| Site | Genera | Species | Larvae collection | | Adult collection | |
|---|---|---|---|---|---|---|
| | | | n | % | n | % |
| **Songon-Agban** | *Aedes* | *Ae. aegypti* | 1,482 | 81.8 | 108 | 52.2 |
| | | *Ae. luteocephalus* | 1 | 0.1 | 0 | 0 |
| | *Culex* | *Cx. nebulosus* | 301 | 16.6 | 44 | 21.3 |
| | | *Cx. quinquefasciatus* | 18 | 1 | 55 | 26.6 |
| | *Mansonia* | *Man. africana* | 3 | 0.2 | 0 | 0 |
| | *Toxorhynchites* | *Toxo. Spp* | 6 | 0.3 | 0 | 0 |
| **Tchancthévè** | *Aedes* | *Ae. aegypti* | 789 | 95.5 | 68 | 69.4 |
| | | *Ae. luteocephalus* | 9 | 1.1 | 2 | 2.0 |
| | *Culex* | *Cx. nebulosus* | 26 | 3.1 | 20 | 20.4 |
| | | *Cx. quinquefasciatus* | 2 | 0.3 | 0 | 0 |
| | *Anopheles* | *An. gambiae* | 0 | 0 | 8 | 8.2 |
| **Kaforo** | *Aedes* | *Ae. aegypti* | 1310 | 68.7 | 13 | 100 |
| | *Culex* | *Cx. quinquefasciatus* | 590 | 31 | 0 | 0 |
| | *Toxorhynchites* | *Toxo. Spp* | 8 | 0.3 | 0 | 0 |

n, number of mosquitoes identified.

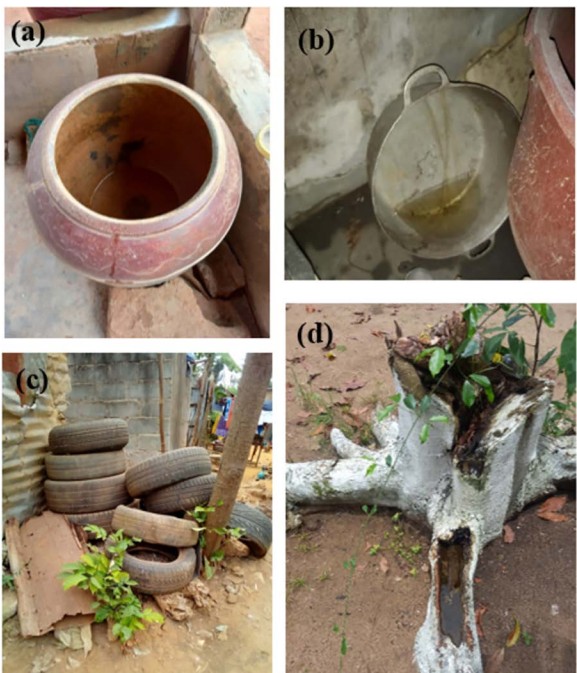

**Fig 2. Potential breeding sites of *Aedes* mosquito sampling in the agricultural areas in Côte d'Ivoire. i.e., a) water storage containers, b) discarded tanks, c) used tires, d) natural containers.**

small water containers (12.3%; 34/276), other tanks (9.8%; 27/276), used tires (9.1%; 25/276) and natural containers (0.4%; 1/276). Of these, 53 breeding sites (19.3%) were found to be infested with *Ae. aegypti*. The positive breeding containers consisted of 38.9% discarded tanks (21/53), 24.1% water storage tanks (13/53), 13.0% small water containers (7/53), 11.1% used tyres (6/53), 11.1% other containers (6/53) and 1.9% natural containers (1/5). Similar to Songon-Agban, discarded tanks were the most frequently positive breeding containers ($\chi^2 = 26.82$; p < 0.001).

In the Kaforo area, a total of 267 potential mosquito breeding sites was identified during a survey involving 66 residential properties. The potential breeding containers were categorised as 57.3% water storage tanks (153/267), 19.5% other tanks (52/267), 13.1% discarded tanks (35/267); 7.5% used tires (20/267) and 2.6% small water containers (7/267). Of the 267 potential breeding sites identified, 55 (20.6%) were found to be infested with *Ae. aegypti* larvae and/or pupae. The positive breeding containers consisted of 38.2% water storage tanks (21/55), 32.7% other containers (18/55), 23.6% discarded tanks (13/55), 3.6% used tires (2/55) and 1.8% small water tanks. In contrast to two other sites, water storage tanks (38.2%) were the most prevalent *Ae. aegypti* breeding site in Kaforo ($\chi^2 = 20.84$; p < 0.001) (Fig 3).

### Homogeneity of Stegomia and pupal indices across the three agricultural areas

The Stegomyia and pupal indices are presented in Table 2. There was no difference between study sites in the Stegomyia indices (HI, CI and BI). House indices of 41.3%, 33.1% and 43.9% were recorded, respectively, in Songon-Agban, Tchancthévè and Kaforo. No significant difference was observed between the three sites ($\chi^2 = 2.70$; p = 0.26). The container indices were 20.9%, 19.6% and 20.6% in the three sites respectively, and there were no significant differences among these sites ($\chi^2 = 0.01$, p = 0.9). Breteau indices were 60.7, 49.5 and 83.3 in Songon-Agban, Tchancthévè and Kaforo, respectively. These BI values are considered very high based on the WHO classification (threshold level >20), suggesting a high risk for dengue transmission in the three sites. The productivity indice per house (PIH) was recorded

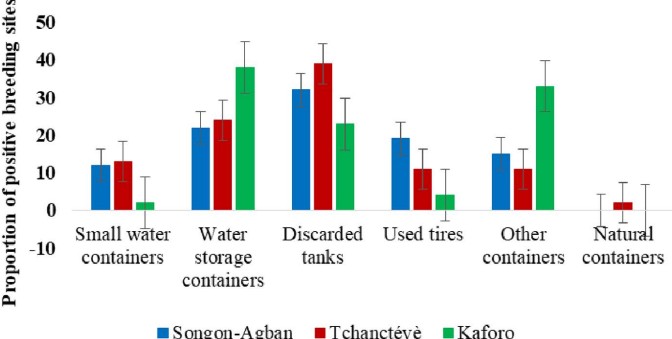

**Fig 3. Proportions of positive breeding sites types in three agricultural areas in Côte d'Ivoire.**

**Table 2. Infestation indices of *Ae aegypti* in three agricultural areas in Côte d'Ivoire.**

| Site | Stegomya Indice | | | | | Pupae Indice | | | | |
|---|---|---|---|---|---|---|---|---|---|---|
| | CI | 95% ci | HI | 95% ci | BI | PIP | 95% ci | PIH | 95% ci |
| **Songon-Abgan** | 20.7 | 16.9 - 24.5 | 41.3 | 33.4 - 49.3 | 60.7 | 0.4 | 0.2 - 0.6 | 2.2 | 1.1 - 3.2 |
| **Tchancthévè** | 18.9 | 14.2 - 23.6 | 32.1 | 23.2 - 41 | 49.5 | 0.2 | 0.1 - 0.23 | 0.8 | 0.2 - 1.5 |
| **Kaforo** | 20.6 | 15.7 - 25.5 | 43.9 | 31.6 - 56.2 | 83.3 | 0.1 | 0.01 - 0.2 | 1.1 | 0.4 - 1.8 |

CI, Container Index; ci, confidence interval; HI, House Index; BI, Breteau Index; PIP, Productivity Index per Person; PIH, Productivity Index per House.

as 2.2, 1.1 and 0.8 pupae per house surveyed, respectively, in Songon-Agban, Tchancthévè and Kaforo. The productivity index per person (PIP) was estimated at 0.4, 0.2 and 0.1 respectively, in Songon-Agban, Tchancthévè and Kaforo. No significant differences were found between these sites (KW = 3.73; p = 0.15).

### Detection of DENV in *Ae. aegypti*

The RT-qPCR was performed on 104 pools containing up to 20 *Ae. aegypti* females (96 pools of adults emerging from larval collections and 8 pools from Prockopack aspiration). In Songon-Agban, 1 pool of *Ae. aegypti* from the larval collection tested positive for DENV with threshold cycle value of 29.67. No positive pools were detected from other sites. The serotyping of the only positive pool detected DENV-3, with threshold cycle value of 34.96. The resulting % Positivity was 3.6% and MIR was 1.9 per 1000 mosquitoes for *Ae. aegypti* in Songon-Agban (Table 3).

### Discussion

The study showed a predominance of *Ae. aegypti* in the domestic and peridomestic breeding sites surveyed, followed by *Cx. quinquefasciatus* and *Cx. nebulosus*. The coexistence of *Ae. aegypti* with these *Culex* species can be explained by

**Table 3. Dengue virus detection in *Ae. aegypti* populations, including dengue serotype identified, percentage of pools testing positive (% Positivity), and minimum infection rate (MIR).**

| Site | Specimens of aspiration analysed | Positive pools/ analysed pools | Specimens of larval prospection analysed | Positive pools/ analysed pools | Serotype DENV | % Positivity | MIR |
|---|---|---|---|---|---|---|---|
| **Songon-Abgan** | 63 (4) | 0/ 4 | 515 (28) | 1/ 28 | DENV-3 | 3.6 | 1.9 |
| **Tchanthévè** | 46 (3) | 0/ 3 | 536 (35) | 0/ 35 | No | 0 | 0 |
| **Kaforo** | 10 (1) | 0/ 1 | 583 (33) | 0/ 33 | No | 0 | 0 |

the fact that they colonise the same types of breeding sites, particularly in rural areas such as discarded tanks, used tires and other containers [47,48].

Our data revealed that *Ae. aegypti* colonized all container types surveyed across the three sites. In Kaforo, the main breeding containers for this species were water storage tanks. The heavy infestation of this category of containers by *Ae. aegypti* is explained by the absence of a water supply system in the houses of this village. The access to drinking water for the local inhabitants depends only on the two water pumps in the village. They store water for drinking and domestic use in clay pots and barrels inside each house, as in many other localities from Côte d'Ivoire [14,15]. This practice is a main risk factor for the presence of *Aedes* vectors in West Africa [16]. Howerver, we found that discarded tanks were the main breeding containers for *Ae. aegypti* in Songon-Agban and Tchancthèvè. Unlike Kaforo, these two villages have permanent access to drinking water. People do not need to store water permanently in containers for domestic use. The main problem in these villages is the lack of management of waste from domestic and agricultural activities, leading to numerous discarded tanks, which provide easy breeding containers for *Ae. aegypti*. These containers can hold water and organic matter for long periods, providing a stable environment for the proliferation of diverse mosquito species like *Ae. aegypti* and *Culex spp.* [49–51]. Discaded containers represented the second category of breeding containers colonised by *Ae. aegypti* in several neighbourhoods of Abidjan [14,15]. The same breeding site was found heavily infested by *Ae. aegypti* recorded in agricultural areas from southeastern Côte d'Ivoire [49]. These results show that the living conditions of these populations facilitate the colonization of containers such as domestic and peridomestic containers in these agricultural areas.

Our results found that natural containers were the least likely to be infested by *Ae. aegypti*. This could be explained by the fact that the study was carried out in an agricultural area. Indeed, the creation of pineapple, cotton and vegetable crops in these areas has primarily required the deforestation to make the land viable for farming purposes. This has led to the destruction of habitats that could provide natural breeding sites for *Ae. aegypti*. In the three study sites, the cultivated plots are on the edges of the village. In addition, farmers have to maintain the plots regularly for insecticide treatments and crop growth. All these conditions create an environment that is not conducive to the creation of natural containers for this species.

The level of transmission risk was estimated using traditional Stegomyia indices (BI, CI and HI) and pupal indices (PIH, PIP). These indices were found to be high and relatively homogeneous in our study sites. All the values of Stegomyia indices suggested high risk for DENV transmission in the three agricultural areas [44,45]. Vector surveillance is recommended by the WHO in dengue-endemic countries. The objective of this study is to provide data on the patterns and geographical distribution of dengue vector populations. Ultimately, this data will be used to inform the management of larval sources, predict transmission risk and evaluate control measures.

However, some studies have questioned the reliability of these indices because they are poorly correlated with the abundance of adult mosquitoes, which should be sampled directly [38,39]. These authors recommend sampling adult mosquitoes directly or indirectly by sampling pupae, the stage just before they emerge as adults. In our study, the productivity indices per person obtained at the tree sites were all bellow the transmission risk threshold (0,5–1,5 pupae per person) [46]. However, in Songon-Agban, PIP was very close to the threshold. This suggested that Songon-Agban may be a higher-risk area for dengue.

Laboratory analyses detected the presence of DENV-3 in *Ae. aegypti* from Songon-Agban through transovarial transmission. Previous studies have shown DENV-2 and -3 infections in *Ae. aegypti* from Abidjan City and rural sites of Côte d'Ivoire [28–30]. However, these studies didn't investigate the potential transmission mechanisms of these viruses from *Ae. aegypti*. Our results indicate that the dengue virus is circulating within the *Ae. aegypti* populations in Côte d'Ivoire, particulary in Songon-Agban although the rate of transovarial transmission of the DENV-3 detected was low. DENV-3 can cause severe cases of dengue and lead to the death of the patient [52–54]. As a result, this arbovirus became of greater concern for the health authorities and is now considered as a major public health issue following the first case of DENV-3 detected in 2008 [7].

The vertical transmission of dengue viruses by mosquitoes has been proposed as a potential mechanism for the persistence of these viruses in mosquito populations [55]. The detection of transovarial transmission of DENV in Songon-Agban could potentially increase the risk of a dengue epidemic in the commune of Songon, including the Songon-Agban site and neighbouring communes. In the same year that the samples were collected, 2022, Côte d'Ivoire was affected by an epidemic of dengue fever. The Institute Pasteur of Côte d'Ivoire reported the circulation of both serotypes 1 and 3 among patients in the Yopougon health district, which encompasses the community of Songon [56]. Considering these results, the detection of transovarial transmission could serve as an early warning sign of a dengue outbreak in Côte d'Ivoire. Given the high risk of dengue fever in Songon-Agban, there is an urgent need to monitor dengue vectors. Monitoring larval habitats and dengue virus circulation in immature *Ae. aegypti* populations could potentially serve as a means of predicting and reducing dengue transmission in the country. However, in-depth studies on the viability of these transovarially infected females in the field are needed to better define the contribution of vertical transmission of DENV to the epidemiology of dengue. For some authors, the importance of vertical transmission in the epidemiology of the disease remains unclear. Numerous studies have been published on this subject [57–59]. These studies suggest that control of vertical transmission is not a sufficient predictor of dengue epidemics. A combination of asymptomatic infection in humans and movement of people are likely to be more important determinants of dengue's persistence [60]. Consequently, strategies to prevent and control the dengue epidemic in Côte d'Ivoire must take these two factors into account, in addition to the transovarial transmission of the virus by the vector.

## Conclusion

This study showed that *Ae. aegypti*, an urban vector of dengue, was the predominant species collected at the three sites. In addition, *Ae. luteocephalus*, a selvatic vector, was found in lower proportion at Songon-Agban and Tchancthévè. Water storage tanks and discarded tanks were the main breeding containers colonised by *Ae. aegypti* in three agricultural areas studied. The high prevalence of these two categories of breeding containers is due to the lack of a water supply system and the poor management of domestic and agricultural waste on these sites. Entomological indices showed that Songon-Agban is a high dengue risk area with significant PIP and the detection of transovarial transmission of DENV-3 in the *Ae. aegypti* population. Monitoring and treating the habitats of *Ae. aegypti* larvae, mainly the water storage tanks and discarded tanks, and monitoring the circulation of dengue virus in immature populations of this species could potentially serve as a means of predicting and reducing dengue transmission in these communes and in country. These prevention and control strategies must also consider cases of asymptomatic infection and the movement of residents due to the high level of commercial activity in this area, which could be a gateway for the spread of DENV-3.

## Supporting information

**S1 Data. List of total number of houses and breeding containers inspected, type of positive breeding containers, number of pupae and inhabitants per house in the three agricultural sites.**
(XLSX)

**S2 Data. List of mosquito samples by the collection method, Detection of dengue virus and serotyping of DENV-1 to -4 determined by reverse transcribed quantitative polymerase chain reaction (rt-qPCR).** The number of threshold cycles in which the dengue virus was detected is indicated in the database.
(XLSX)

## Acknowledgments

The authors wish to acknowledge the contributions of Mr Bassely Edmond Charles, Ms. Kouadio Affoué Mireille Nadia, Zokou Aleigne Elizabeth and Coulibaly Assiata for assistance in field activities. We thank Drs Kamo Emilie, Kouakou Luc-Venance and Diobo N'Guessan Fidèle, Mr Gueu Ghislain, Ms Koffi Ahou Melissa and the rest of the personel of the

Epidemic Viruses Department of the Institut Pasteur de Côte d'Ivoire for their support in laboratory activities. We would also like to thank the health authorities, local authorities and inhabitants of the study areas, as well as the entomological team who worked on this study. This research was integrated into the RESIS-ARBO project.

## Author contributions

**Conceptualization:** Edwige M.-A. Y. Kadjo, Maurice A. Adja.

**Data curation:** Edwige M.-A. Y. Kadjo, Sylla Yahaya, Valery E. Adjogoua.

**Formal analysis:** Edwige M.-A. Y. Kadjo, Sylla Yahaya, Négnorogo Guindo-Coulibaly, François D. Traoré, Maurice A. Adja.

**Funding acquisition:** Maurice A. Adja.

**Investigation:** Edwige M.-A. Y. Kadjo, Négnorogo Guindo-Coulibaly, Danielle D. Zoh, Fabrice K. Assouho, Mardoché R.K. Azongnibo, Stéphane D.M. Kpan.

**Methodology:** Edwige M.-A. Y. Kadjo, Sylla Yahaya, François D. Traoré, Fabrice K. Assouho, Mardoché R.K. Azongnibo, Stéphane D.M. Kpan, Maurice A. Adja.

**Project administration:** Maurice A. Adja.

**Resources:** Valery E. Adjogoua, Fabrice Chandre, Maurice A. Adja.

**Software:** Edwige M.-A. Y. Kadjo, Sylla Yahaya, Négnorogo Guindo-Coulibaly, Danielle D. Zoh, Fabrice K. Assouho, Mardoché R.K. Azongnibo.

**Supervision:** Valery E. Adjogoua, Ahoua Yapi, Fabrice Chandre, Maurice A. Adja.

**Validation:** Valery E. Adjogoua, Ahoua Yapi, Fabrice Chandre, Maurice A. Adja.

**Visualization:** Edwige M.-A. Y. Kadjo, Négnorogo Guindo-Coulibaly, Valery E. Adjogoua, Fabrice Chandre, Maurice A. Adja.

**Writing – original draft:** Edwige M.-A. Y. Kadjo.

**Writing – review & editing:** Edwige M.-A. Y. Kadjo, Sylla Yahaya, Négnorogo Guindo-Coulibaly, Danielle D. Zoh, François D. Traoré, Fabrice K. Assouho, Mardoché R.K. Azongnibo, Stéphane D.M. Kpan, Ahoua Yapi, Fabrice Chandre, Maurice A. Adja.

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
