## [Decision Letter · Decision Letter 0]

Dear Student KADJO,

Thank you very much for submitting your manuscript "Abidjan, December 18th 2023

Transovarial transmission of dengue virus in Aedes aegypti from three agricultural areas with high-risk epidemic in Côte d'Ivoire." for consideration at PLOS Neglected Tropical Diseases. As with all papers reviewed by the journal, your manuscript was reviewed by members of the editorial board and by several independent reviewers. In light of the reviews (below this email), we would like to invite the resubmission of a significantly-revised version that takes into account the reviewers' comments. 

The study provides important information on Ae. aegypti ecology in agricultural settings, but there are several limitations and issues that need to be addressed to warrant its publication. I am particularly worried about the excessive attention towards transovarial transmission (starting with the title) and dengue epidemiology given the limited data on this aspect of the study. The authors should carefully address each issue raised by the Reviewers and consider modifying the focus of the paper to mosquito ecology and include transovarial transmission as a secondary outcome. Ae. aegypti ecology certainly matters to dengue epidemiology, so the authors can discuss that in a broader context. Details about the molecular assays to detect DENV should be provided (including Ct values).

We cannot make any decision about publication until we have seen the revised manuscript and your response to the reviewers' comments. Your revised manuscript is also likely to be sent to reviewers for further evaluation.

Sincerely,

Tereza Magalhaes, Ph.D.

Academic Editor

Audrey Lenhart

Section Editor

The study provides important information on Ae. aegypti ecology in agricultural settings, but there are several limitations and issues that need to be addressed to warrant its publication. I am particularly worried about the excessive attention towards transovarial transmission (starting with the title) and dengue epidemiology given the limited data on this aspect of the study. The authors should carefully address each issue raised by the Reviewers and consider modifying the focus of the paper to mosquito ecology and include transovarial transmission as a secondary outcome. Ae. aegypti ecology certainly matters to dengue epidemiology, so the authors can discuss that in a broader context. Details about the molecular assays to detect DENV should be provided (including Ct values).

Reviewer's Responses to Questions

**Key Review Criteria Required for Acceptance?**

**Methods**

-Are the objectives of the study clearly articulated with a clear testable hypothesis stated?

-Is the study design appropriate to address the stated objectives?

-Is the population clearly described and appropriate for the hypothesis being tested?

-Is the sample size sufficient to ensure adequate power to address the hypothesis being tested?

-Were correct statistical analysis used to support conclusions?

-Are there concerns about ethical or regulatory requirements being met?

Reviewer #1: (No Response)

Reviewer #2: Overall, this study is exploratory with the broad goal of improving understanding of the dengue situation in Côte d'Ivoire. 

The objectives of the study ("The present study aims to assess the levels of epidemic risk and of circulation to DENV in Ae. aegypti from three agricultural areas of Côte d'Ivoire.") are overly broad and do not address a clear question. This stated objective is not in line with the title “Transovarial transmission of dengue virus in Aedes aegypti from three agricultural areas with high-risk epidemic in Côte d'Ivoire.” Clear objectives and hypotheses for statistical tests are needed to assess the appropriateness of the experimental design.

Sampling effort should be reported. How often was each house sampled? Only once? 

How were larvae collected? Were all larvae in a container collected?

Data analysis: what is the justification for performing the statistical tests described? What are the hypotheses being tested with these tests and why do they matter?

Reviewer #3: The objetive is clearly articulated. It does not state an hypothesis, which shouldn´t be required considering it´s a descriptive work which “aims to assess the levels of epidemic risk and of circulation to DENV in Ae. aegypti from three agricultural areas of Côte d'Ivoire.”. The sampling design is appropriate to address the stated objetive. The sample size is sufficient and statistical analysis were adequate

**Results**

-Does the analysis presented match the analysis plan?

-Are the results clearly and completely presented?

-Are the figures (Tables, Images) of sufficient quality for clarity?

Reviewer #1: For Table 1, it might be clearer to use the column heading “Adult Collection” instead of “Aspiration.” For Table 2, “confidence interval” should be referred to as “ci” consistently throughout the table column headings. For Table 3, the column headings run together, and it is difficult to understand what each column represents. For Figure 2, it would be beneficial to explain which breeding site type (categories in Figure 3) is represented by each picture. For example, is picture (a) an example of a small water container or a water storage container? Also, in Figure 3, the vertical axis should read “Proportion of Positive Breeding sites.”

Reviewer #2: Table 3: pool numbers are 28, 35, and 35, total = 98. In the text, total is mentioned as 96.

Reviewer #3: Most of the results are well and completely presented (however, English grammar and spelling need thorough revision). Figures and Tables are adequate and clear.

There is no actual reference to the number of containers examined per container type, then it is not clear whether a higher proportion of a positive container type shows potential larval habitat availability (eg. there are more discarded tanks in Songon-Agban than other container categories) or preference (i.e. discarded tanks are prefered over other container types as larval habitat). If there is a preference, then the proportion of positives for a given container type should be higher than expected by chance (at least higher than the overall positive proportion). This information (the actual number examined per category, or the proportion positive per category) should be included, specially considering that certain container types are later discussed as “the most prevalent and the most productive breeding sites”.

**Conclusions**

-Are the conclusions supported by the data presented?

-Are the limitations of analysis clearly described?

-Do the authors discuss how these data can be helpful to advance our understanding of the topic under study?

-Is public health relevance addressed?

Reviewer #1: This study provides useful information regarding breeding site preferences for Aedes aegypti populations in these agricultural communities. Some studies suggest that dengue epidemic risk is associated with high Stegomyia indices, but other studies have found no correlation (Example: Garjito et al 2020). It might be useful for the discussion to cite literature describing both viewpoints. 

The authors state in the discussion that transovarial transmission is the overwintering mechanism by which the virus is maintained at low rates during adverse environmental conditions. There are other theories about how dengue viruses persist, including asymptomatic infection in humans and the movement of viremic people (Grunnill and Boots 2016). The discussion would benefit with a more thorough review of the literature on this subject.

Reviewer #2: Is any information available on dengue infection in humans at or near the study sites? The paragraph at L326 would be a good place to discuss these if so.

Some conclusions are overstated. Please be sure conclusions are in line with the results and study limitations. For example: L336: “This result clearly confirm that the risk of dengue epidemics

is higher at this site” – this statement is based on the detection of DENV from one positive pool out of 28 pools, and 0 positive pools out of 35 pools at each of the other sites. Based on this, we cannot clearly confirm that risk of dengue epidemic is higher at this site. Choice of words is very important as well, as these data do not predict anything about epidemics of dengue at these sites.

There are substantial limitations of this study that are not discussed. Discussion of these limitations is necessary to properly interpret the results. Limitations that should be discussed are listed below: 

One pool of Ae. aegypti reared from immature collections, out of 98 tested (or 96?), was positive for DENV. In the discussion, this result is presented as the rate of transovarial transmission, and it should be mentioned that this rate is calculated from only one detection. Similarly, it is misleading to present 3.6% as the rate of transovarial transmission because that calculation, 1/28, is based on detection of DENV in pools of 20 mosquitoes. Because mosquitoes are pooled, the true rate of transovarial transmission in this population, based on these data, is likely substantially lower, as low as 0.17% - a very different number.

Mosquito sampling took place over a limited timeframe, and this should be mentioned in the discussion so that the results are properly framed. 

Importantly, limitations of the Stegomyia indices need to be discussed. Previous work on the lack of correlation between indices and dengue risk should be addressed in the discussion, e.g., Garjito et al. 2020, Stegomyia Indices and Risk of Dengue Transmission: A Lack of Correlation, Frontiers in Public Health, 8: 2020-00328. If any data or information is available about past dengue transmission at or near the study sites, that would be important information to include where the implications of Stegomyia index results are discussed.

Reviewer #3: The conclusions are partly supported by the data presented. 

In order to support the first statement of the conclusion, i.e. “that Ae. aegypti preferentially colonized two categories of breeding sites…”, a reference to the number of containers examined per category is needed. Results showed the proportion of positive containers per category, but not for a given category the proportion of container that were positive (see comments on results). This should be easily revised and corrected by the authors.

The authors discuss the results in terms of their implications on dengue transmission risk; limitations of analysis and public health relevance are addressed

- Out of curiosity, even though the study was carried out in a rural area, there is a very small proportion of positive natural containers, is this likely a sampling bias towards artificial containers?

- The detection of transovarial DENV transmission may be a useful early warning sign of an outbreak. Were mosquito control activities increased following your survey? What was the dengue situation following the detection of the transovarial transmission? Were human cases reported? Were they DENV III (or at least at the beginning of the outbreak)? It would be useful if you can add some of this information or at least discuss it.

**Editorial and Data Presentation Modifications?**

Reviewer #1: This manuscript would benefit from additional copyediting throughout to improve readability. 

In addition, a few minor editing suggestions:

1. p.1, line 2 Larval and adult collections were carried out… 

2. p.4, line 67 Spell out World Health Organization

3. p.4, line 73 Abbreviate World Health Organization

4. p.8, line 167 Provide Prokopack manufacturer information in parentheses

Reviewer #2: There are numerous presentation issues throughout the manuscript and the authors should carefully revise the manuscript to address these. These issues include numerous typographical and grammatical errors, formatting inconsistencies, and unclear and inaccurate wordings. Please carefully edit the document to correct these and to improve flow and clarity.

Reviewer #3: The work seems based on solid fieldwork, however it needs a thorough revision for grammar and spelling (please see just a few examples below).

Line 78: “waste of civilization”, you probably meant “urban waste”.

Line 219, should be < 3%

Throughout the text, please note that it should be “species” (species is both singular and plural, specie is not correct)

There are several typos along the manuscript that should be corrected, for example (to name a few): “mousquitoes” (should be mosquitoes, line 190); “detecteion” (line 201, should be detection), “theses” (line 233, these), line 256 and 257 “contenairs” (containers), “wich” (which)

Line 236, omit “of” in “78.9% of Aedes, 20.9% of Culex, 0.4% of Toxorhynchites and 0.1% of Anopheles”

Line 238 “too vector of dengue” should be “vector of dengue, too”

<b>Summary and General Comments

Reviewer #1: In this manuscript, the authors describe field studies conducted in three agricultural areas in Cote d’Ivoire to investigate Aedes aegypti populations and the risk for dengue virus activity. Immature mosquitoes (larvae) and adult mosquitoes were collected and tested for dengue viruses using a real-time RT-PCR assay to investigate vertical transmission (VT) and to determine infection rates. Entomological surveys, including both adult and pupal indices, were conducted to estimate dengue transmission risk. Studies evaluating VT are important as information is lacking about how mosquito-borne arboviruses persist trans-seasonally. Vertical transmission of dengue viruses has been previously reported, but the epidemiological significance remains controversial. The findings from this study document the occurrence of VT of dengue in Aedes aegypti mosquitoes in Cote d’Ivoire. This manuscript also provides new insight regarding potential epidemic risk for dengue transmission based on entomological indices in these agricultural areas.

Reviewer #2: The manuscript "Transovarial transmission of dengue virus in Aedes aegypti from three agricultural areas with high-risk epidemic in Côte d'Ivoire" by Kadjo et al. is an exploratory investigation of Aedes aegypti occurrence and larval habitats, and dengue virus infection in that mosquito. This study provides useful data on Aedes aegypti and dengue virus in Côte d'Ivoire, but substantial revisions are needed. The study lacks a centralized theme and set of hypotheses that are tested to answer a question. The title frames this as an investigation of transovarial transmission of DENV, but this topic is a minor component of the overall data that were collected. No clear hypotheses or objectives are presented, and there are numerous typological, grammatical, and consistency errors, to the extent that it is difficult to understand what is being communicated. English language editing would be helpful for some of these, but the authors should also work to ensure consistency in terminologies, proper formatting, and corrections to minor errors. Editing the manuscript for flow and clarity would be beneficial as well.

Reviewer #3: The manuscript assessed the container mosquito fauna from three agricultural areas in Côte d’Ivoire, tested Aedes aegypti for DENV and, most importantly, provide evidence of transovarial transmission of the DENV III, based on the proportion of immature mosquitoes collected in the field that were positive. Moreover, they identify the relative importance of different container types as larval habitat of Ae. aegypti and estimate risk indices (classical Stegomyia indexes plus pupal indexes). More productive container types differed between areas, depending on the villages having or not permanent access to drinking water. This information should be especially useful for policy makers. 

Even though transovarial transmission of dengue virus by Aedes aegypti has been reported in the literature, most dengue control interventions aim at interrupting horizontal transmission. Considering that vertical transmission may be an important virus maintenance mechanism, reports of the actual occurrence and rates of transovarial transmission (this work) in different regions are a valuable resource to better understand the ecoepidemiology of DENV and help improve prevention and/or control strategies. 

The manuscript needs a thorough grammar and spelling revision, and some clarifications /additional information regarding the results reported (see comments on the Results section). The proportion of positive containers per category to the number of containers examined in that particular category should be provided in order to support the first statement of the conclusion, i.e. “that Ae. aegypti preferentially colonized two categories of breeding sites…”. This should be easily revised and corrected by the authors.

It would be useful if the authors could add or discuss more actual information on the DENV situation following the detection of transovarial DENV transmission in the region, since it can be a useful early warning sign of an outbreak. Were mosquito control activities increased following your survey? What was the dengue situation following the detection of the transovarial transmission? Were human cases reported? Were they DENV III (or at least at the beginning of the outbreak)?

PLOS authors have the option to publish the peer review history of their article (what does this mean? ). If published, this will include your full peer review and any attached files.

**Do you want your identity to be public for this peer review?** For information about this choice, including consent withdrawal, please see our Privacy Policy .

Reviewer #1: No

Reviewer #2: No

Reviewer #3: No
---

## [Decision Letter · Decision Letter 1]

Dear Student KADJO,

Thank you very much for submitting your manuscript "Ecology and transmission risk of dengue by Ae. aegypti in three leading agricultural areas of Côte d’Ivoire" for consideration at PLOS Neglected Tropical Diseases. As with all papers reviewed by the journal, your manuscript was reviewed by members of the editorial board and by several independent reviewers. In light of the reviews (below this email), we would like to invite the resubmission of a significantly-revised version that takes into account the reviewers' comments. 

The reviewers acknowledged significant improvements in the manuscript but highlighted several important issues that still need to be addressed. Both reviewers emphasized that the language requires a thorough revision. One reviewer also noted that several details, not just related to grammar but also to the coherence among methodology, results, and discussion, require revision.

Additionally, all reviewers pointed out that the transovarial transmission rate (TOT) was likely calculated incorrectly. Please refer to their comments for more details. The work cited by the authors shows that TOT was calculated as MIR per 1,000 mosquitoes, which differs from the method described by the authors. This further supports the need for revision, as the TOT value currently presented in the manuscript is an overestimation.

Please change the DENV serotype description from Roman numerals to Arabic numerals, as Roman numerals are typically used to refer to DENV genotypes.

We cannot make any decision about publication until we have seen the revised manuscript and your response to the reviewers' comments. Your revised manuscript is also likely to be sent to reviewers for further evaluation.

Sincerely,

Tereza Magalhaes, Ph.D.

Academic Editor

Audrey Lenhart

Section Editor

The reviewers acknowledged significant improvements in the manuscript but highlighted several important issues that still need to be addressed. Both reviewers emphasized that the language requires a thorough revision. One reviewer also noted that several details, not just related to grammar but also to the coherence among methodology, results, and discussion, require revision.

Additionally, all reviewers pointed out that the transovarial transmission rate (TOT) was likely calculated incorrectly. Please refer to their comments for more details. The work cited by the authors shows that TOT was calculated as MIR per 1,000 mosquitoes, which differs from the method described by the authors. This further supports the need for revision, as the TOT value currently presented in the manuscript is an overestimation.

Please change the DENV serotype description from Roman numerals to Arabic numerals, as Roman numerals are typically used to refer to DENV genotypes.

Reviewer's Responses to Questions

**Key Review Criteria Required for Acceptance?**

**Methods**

-Are the objectives of the study clearly articulated with a clear testable hypothesis stated?

-Is the study design appropriate to address the stated objectives?

-Is the population clearly described and appropriate for the hypothesis being tested?

-Is the sample size sufficient to ensure adequate power to address the hypothesis being tested?

-Were correct statistical analysis used to support conclusions?

-Are there concerns about ethical or regulatory requirements being met?

Reviewer #1: (No Response)

Reviewer #2: How were Toxorhynchites larvae reared? Were they reared alongside other collected larvae, and did they consume other collected larvae? Did this affect the results?

L161: Were all larvae present in an inspected container collected? Could bias have been introduced by the volume of water in the container or the type of container? For example, if the volume was large, was it possible to collect all the larvae? Or, if the container was a natural container, could the presence of debris have made is challenging to collect all larvae that were present? Please mention these factors in the Methods and discuss in the Discussion how they may have affected larval habitat use results.

L188: should this read "up to 20..."?

**Results**

-Does the analysis presented match the analysis plan?

-Are the results clearly and completely presented?

-Are the figures (Tables, Images) of sufficient quality for clarity?

Reviewer #1: For Table 3, the wording needs to be changed to reflect % positivity instead of transovarial transmission rate. The term "transovarial transmission rate" refers to the % of transovarially infected females that pass the virus to their progeny. Consider editing Line 306 to say something like: "Evidence supporting transovarial transmission of dengue virus in Aedes aegypti populations, including dengue serotype identified, percentage of pools testing positive (% Positivity), and minimum infection rate (MIR). " In the table, change "TOR" to "% Positivity." Additional edits should be made throughout the paper to reflect that 3.6% indicates % positivity and is not a TOT rate.

Reviewer #2: L31/L396: should read "Aedes aegypti was the predominant species in the samples collected at the three sites" or something like "Aedes aegypti was predominant among the fauna of the sites based on sampling of containers and aspirations." It cannot be concluded that Aedes aegypti is the dominant or most abundant species within the faunas of the sites because of sampling bias. 

L35: Please include that DENIII was detected in one pool/96.

**Conclusions**

-Are the conclusions supported by the data presented?

-Are the limitations of analysis clearly described?

-Do the authors discuss how these data can be helpful to advance our understanding of the topic under study?

-Is public health relevance addressed?

Reviewer #1: (No Response)

Reviewer #2: L368: Please reword for accuracy - DENV is not circulating among larvae as it is not transmitted from larva to larva.

**Editorial and Data Presentation Modifications?**

Reviewer #1: The manuscript needs some additional editing to address spelling and grammar issues.

Reviewer #2: As I describe in the general comments, the remaining major issue in this manuscript is the abundance of minor errors. The authors need to revise the manuscript with aggressive attention to detail to fix the various consistency, grammatical, typographical and wording errors.

**Summary and General Comments**

Reviewer #1: The authors made an effort to address the reviewer comments and questions.

Reviewer #2: Overall, the authors have done a nice job of revising the manuscript according to the reviewers' comments and the revised manuscript is a substantially improved. The one remaining major item to be addressed is that the manuscript still needs to be revised and detailed to fix formatting, consistency, grammatical, typographical and flow errors. I have recommended a major revision because of the quantity of these minor errors. Please edit the manuscript, with attention to detail, to address these issues - some of these are related to English language standards, but there are many instances of formatting inconsistencies and typos (e.g., italicization of species names or words next to species names, misspelled words, inconsistent capitalization, Côte d'Ivoire vs Ivory Coast, dengue 3 vs. DENIII, dengue virus vs. DENV, only define abbreviations once at the first mention of the phrase, after an abbreviation is defined, use the abbreviation unless at the beginning of a sentence, etc.).

PLOS authors have the option to publish the peer review history of their article (what does this mean? ). If published, this will include your full peer review and any attached files.

**Do you want your identity to be public for this peer review?** For information about this choice, including consent withdrawal, please see our Privacy Policy .

Reviewer #1: No

Reviewer #2: No
---

## [Editor Report · Decision Letter 2]

PNTD-D-23-01622R2Ecology and transmission risk of dengue by Ae. aegypti in three leading agricultural areas of Côte d’IvoirePLOS Neglected Tropical Diseases Dear Dr. KADJO, Thank you for submitting your manuscript to PLOS Neglected Tropical Diseases. After careful consideration, we feel that it has merit but does not fully meet PLOS Neglected Tropical Diseases's publication criteria as it currently stands. Therefore, we invite you to submit a revised version of the manuscript that addresses the points raised during the review process. Please submit your revised manuscript within 30 days Dec 27 2024 11:59PM. If you will need more time than this to complete your revisions, please reply to this message or contact the journal office at plosntds@plos.org. Please include the following items when submitting your revised manuscript: * A rebuttal letter that responds to each point raised by the editor and reviewer(s). You should upload this letter as a separate file labeled 'Response to Reviewers '. This file does not need to include responses to any formatting updates and technical items listed in the 'Journal Requirements' section below. * A marked-up copy of your manuscript that highlights changes made to the original version. You should upload this as a separate file labeled 'Revised Manuscript with Track Changes '. * An unmarked version of your revised paper without tracked changes. You should upload this as a separate file labeled 'Manuscript '. If you would like to make changes to your financial disclosure, competing interests statement, or data availability statement, please make these updates within the submission form at the time of resubmission. Guidelines for resubmitting your figure files are available below the reviewer comments at the end of this letter. We look forward to receiving your revised manuscript. Kind regards,Tereza Magalhaes, Ph.D.Academic EditorPLOS Neglected Tropical Diseases Audrey LenhartSection EditorPLOS Neglected Tropical Diseases

Shaden Kamhawi

co-Editor-in-Chief

Paul Brindley

co-Editor-in-Chief

**Additional Editor Comments :** Please change the phrasing "The epidemiological impact of vertical transmission of dengue virus is still poorly understood but it is important for the maintenance of the virus in mosquito populations during interepidemic periods [21,22]" by changing "is important" to "may be important".

Please do not use "nymph" for mosquitoes - make sure to change that throughout the ms.

Please conduct an English review as requested by the Reviewers and Editor and go over all requested changes again to make sure you have covered all points. **Journal Requirements:**

At this stage, the following Authors/Authors require contributions: YAPO Marie-Ange Edwige KADJO, Sylla Yahaya, Négnorogo Guindo-Coulibaly, Valery Edgard Adjogoua, Dounin Danielle Zoh, Dipomin François Traoré, Konan Fabrice Assouho, Konan Rodolphe Mardoché Azongnibo, Mintokapieu Didier Stéphane Kpan, Ahoua Yapi, Fabrice Chandre, and Akré Maurice Adja. Please ensure that the full contributions of each author are acknowledged in the "Add/Edit/Remove Authors" section of our submission form.

2) We have noticed that you have uploaded Supporting Information files, but you have not included a list of legends. Please add a full list of legends for your Supporting Information files after the references list.

3) We note that your Data Availability Statement is currently as follows: "The database used to obtain the results presented in this manuscript has been downloaded as supporting information.". Please confirm at this time whether or not your submission contains all raw data required to replicate the results of your study. Authors must share the “minimal data set” for their submission. PLOS defines the minimal data set to consist of the data required to replicate all study findings reported in the article, as well as related metadata and methods (https://journals.plos.org/plosone/s/data-availability#loc-minimal-data-set-definition).

4) Please amend your detailed Financial Disclosure statement. This is published with the article. It must therefore be completed in full sentences and contain the exact wording you wish to be published.

1) State the initials, alongside each funding source, of each author to receive each grant. For example: "This work was supported by the National Institutes of Health (####### to AM; ###### to CJ) and the National Science Foundation (###### to AM).".

---

## [Editor Report · Decision Letter 3]

PNTD-D-23-01622R3Ecology and transmission risk of dengue by Ae. aegypti in three leading agricultural areas of Côte d’IvoirePLOS Neglected Tropical Diseases Dear Dr. KADJO, Thank you for submitting your manuscript to PLOS Neglected Tropical Diseases. After careful consideration, we feel that it has merit but does not fully meet PLOS Neglected Tropical Diseases's publication criteria as it currently stands. Therefore, we invite you to submit a revised version of the manuscript that addresses the points raised during the review process. Please submit your revised manuscript within 30 days Apr 05 2025 11:59PM. If you will need more time than this to complete your revisions, please reply to this message or contact the journal office at plosntds@plos.org. Please include the following items when submitting your revised manuscript: * A rebuttal letter that responds to each point raised by the editor and reviewer(s). You should upload this letter as a separate file labeled 'Response to Reviewers '. This file does not need to include responses to any formatting updates and technical items listed in the 'Journal Requirements' section below. * A marked-up copy of your manuscript that highlights changes made to the original version. You should upload this as a separate file labeled 'Revised Manuscript with Track Changes '. * An unmarked version of your revised paper without tracked changes. You should upload this as a separate file labeled 'Manuscript '.

 We look forward to receiving your revised manuscript. Kind regards, Tereza Magalhaes, Ph.D.Academic EditorPLOS Neglected Tropical Diseases

Audrey Lenhart

Section EditorPLOS Neglected Tropical Diseases

Shaden Kamhawi

co-Editor-in-Chief

Paul Brindley

co-Editor-in-Chief

**Additional Editor Comments:** Please resubmit a change-tracked document showing all modifications made to the most recent version of the manuscript. The highlighted sections in yellow do not correspond to all the changes, and a track-change document is required for the editorial team to properly evaluate the new version. The editorial team will not review the new version unless it is submitted with tracked changes (please use the Track Changes tool in Word).

Additionally, please remove the sentence stating that secondary infections are more severe with specific dengue serotypes, as severe disease also occurs in primary infections, and secondary infection can be a risk for severe disease across all serotypes. **Journal Requirements:**

Please ensure that the funders and grant numbers match between the Financial Disclosure field and the Funding Information tab in your submission form. Note that the funders must be provided in the same order in both places as well.

**Reviewers' comments:** **Figure resubmission:** While revising your submission, please upload your figure files to the Preflight Analysis and Conversion Engine (PACE) digital diagnostic tool, https://pacev2.apexcovantage.com/. PACE helps ensure that figures meet PLOS requirements. To use PACE, you must first register as a user. Registration is free. Then, login and navigate to the UPLOAD tab, where you will find detailed instructions on how to use the tool. If you encounter any issues or have any questions when using PACE, please email PLOS at figures@plos.org. Please note that Supporting Information files do not need this step. If there are other versions of figure files still present in your submission file inventory at resubmission, please replace them with the PACE-processed versions. **Reproducibility:** To enhance the reproducibility of your results, we recommend that authors of applicable studies deposit laboratory protocols in protocols.io, where a protocol can be assigned its own identifier (DOI) such that it can be cited independently in the future. Additionally, PLOS ONE offers an option to publish peer-reviewed clinical study protocols. Read more information on sharing protocols at https://plos.org/protocols?utm_medium=editorial-email&utm_source=authorletters&utm_campaign=protocols

---

## [Editor Report · Decision Letter 4]

PNTD-D-23-01622R4Ecology and transmission risk of dengue by Ae. aegypti in three leading agricultural areas of Côte d’IvoirePLOS Neglected Tropical Diseases  Dear Dr. KADJO, Thank you for submitting your manuscript to PLOS Neglected Tropical Diseases. After careful consideration, we feel that it has merit but does not fully meet PLOS Neglected Tropical Diseases's publication criteria as it currently stands. Therefore, we invite you to submit a revised version of the manuscript that addresses the points raised during the review process. Please submit your revised manuscript within 30 days May 19 2025 11:59PM. If you will need more time than this to complete your revisions, please reply to this message or contact the journal office at plosntds@plos.org. Please include the following items when submitting your revised manuscript: * A rebuttal letter that responds to each point raised by the editor and reviewer(s). You should upload this letter as a separate file labeled 'Response to Reviewers '. This file does not need to include responses to any formatting updates and technical items listed in the 'Journal Requirements' section below. * A marked-up copy of your manuscript that highlights changes made to the original version. You should upload this as a separate file labeled 'Revised Manuscript with Track Changes '. * An unmarked version of your revised paper without tracked changes. You should upload this as a separate file labeled 'Manuscript '. If you would like to make changes to your financial disclosure, competing interests statement, or data availability statement, please make these updates within the submission form at the time of resubmission. Guidelines for resubmitting your figure files are available below the reviewer comments at the end of this letter. We look forward to receiving your revised manuscript. Kind regards, Audrey LenhartSection EditorPLOS Neglected Tropical Diseases Audrey LenhartSection EditorPLOS Neglected Tropical Diseases

Shaden Kamhawi

co-Editor-in-Chief

Paul Brindley

co-Editor-in-Chief

**Additional Editor Comments:** Thank you for your attention to addressing the previous feedback from the reviewers and the Academic Editor. The manuscript is much improved, and it addresses an area of Aedes aegypti ecology in Africa about which little is known. I am attaching my feedback, which includes some significant suggested editing of the English and general grammar, and there are some additional minor points for clarification that I have included as comments in the attached version of the manuscript.**Reviewers' comments:****Figure resubmission:** While revising your submission, please upload your figure files to the Preflight Analysis and Conversion Engine (PACE) digital diagnostic tool, https://pacev2.apexcovantage.com/. PACE helps ensure that figures meet PLOS requirements. To use PACE, you must first register as a user. Registration is free. Then, login and navigate to the UPLOAD tab, where you will find detailed instructions on how to use the tool. If you encounter any issues or have any questions when using PACE, please email PLOS at figures@plos.org. Please note that Supporting Information files do not need this step. If there are other versions of figure files still present in your submission file inventory at resubmission, please replace them with the PACE-processed versions. **Reproducibility:** To enhance the reproducibility of your results, we recommend that authors of applicable studies deposit laboratory protocols in protocols.io, where a protocol can be assigned its own identifier (DOI) such that it can be cited independently in the future. Additionally, PLOS ONE offers an option to publish peer-reviewed clinical study protocols. Read more information on sharing protocols at https://plos.org/protocols?utm_medium=editorial-email&utm_source=authorletters&utm_campaign=protocols

---

## [Editor Report · Decision Letter 5]

Dear PhD KADJO,

We are pleased to inform you that your manuscript 'Aedes aegypti ecology and dengue infection in three agricultural areas of Côte d’Ivoire' has been provisionally accepted for publication in PLOS Neglected Tropical Diseases.

Best regards,

Audrey Lenhart

Section Editor

Audrey Lenhart

Section Editor

Shaden Kamhawi

co-Editor-in-Chief

Paul Brindley

co-Editor-in-Chief

---

## [Editor Report · Acceptance letter]

Dear PhD KADJO,

We are delighted to inform you that your manuscript, "Aedes aegypti ecology and dengue infection in three agricultural areas of Côte d’Ivoire," has been formally accepted for publication in PLOS Neglected Tropical Diseases.

Best regards,

Shaden Kamhawi

co-Editor-in-Chief

Paul Brindley

co-Editor-in-Chief
